# Optical emission near a high-impedance mirror

Majid Esfandyarpour[1], Alberto G. Curto [1,2], Pieter G. Kik[1,3], Nader Engheta [4] & Mark L. Brongersma[1]

Solid state light emitters rely on metallic contacts with a high sheet-conductivity for effective charge injection. Unfortunately, such contacts also support surface plasmon polariton and lossy wave excitations that dissipate optical energy into the metal and limit the external quantum efficiency. Here, inspired by the concept of radio-frequency high-impedance surfaces and their use in conformal antennas we illustrate how electrodes can be nanopatterned to simultaneously provide a high DC electrical conductivity and high-impedance at optical frequencies. Such electrodes do not support SPPs across the visible spectrum and greatly suppress dissipative losses while facilitating a desirable Lambertian emission profile. We verify this concept by studying the emission enhancement and photoluminescence lifetime for a dye emitter layer deposited on the electrodes.

[1] Geballe Laboratory for Advanced Materials, Stanford University, 476 Lomita Mall, Stanford, California 94305, USA. [2] Department of Applied Physics and Institute for Photonic Integration, Eindhoven University of Technology, 5600 MB Eindhoven, The Netherlands. [3] CREOL, The College of Optics and Photonics, University of Central Florida, Florida 32816, USA. [4] Department of Electrical and Systems Engineering, University of Pennsylvania, Philadelphia, PA 19104, USA. Correspondence and requests for materials should be addressed to M.L.B. (email: Brongersma@stanford.edu)

The development of high-efficiency and high-power solid-state light emitters has led to their dramatically increased use in lighting and display applications. Yet there are still significant opportunities to further boost performance. In particular, organic light-emitting diodes exhibit a low external quantum efficiency (EQE), which can largely be attributed to the excitation of dissipative optical modes in the metallic electrode[1,2]. These can be classified into guided surface plasmon polariton (SPP) modes and high-spatial-frequency modes termed lossy waves (LWs)[3,4] Their relative importance depends on the emitter-to-metal spacing (Supplementary Fig. 1, Supplementary Note 1). The electrodes in light emitting devices perform critical functions by facilitating carrier injection and serving as a mirror that redirects photons to the emitting surface. As such, they cannot be omitted from the device stack and these optical loss channels seem unavoidable.

Currently, optical energy dissipation in metallic electrodes is minimized by inserting a dielectric spacer layer between the metal and emitter layers. However, the use of such spacers increases the device's contact resistance and the probability of trapping light inside the high-index light emitting materials via total internal reflection[5]. Significant progress has been made in light extraction from light emitting devices (LEDs)[6–10]. Different nanophotonic structures and metamaterials have been used to modify the emission of quantum emitters[11–13]. Alternatively, researchers have attempted to reduce the SPP mode loss contribution by using wavelength-scale periodic gratings in the electrode that decouple excited SPPs into free-space radiation[14–17]. Although these type of structures can out-couple SPPs very efficiently, they give rise to a highly-directional and wavelength-dependent emission[17], which is undesirable in many display and lighting applications. In light of these drawbacks, it would be highly desirable to tackle the dissipative loss in the metal at its root and reduce the coupling of emitters to SPPs and LWs in the electrode without the need for spacer layers and complex light-extraction schemes.

Our approach to solve this problem is inspired by works originally performed in radio frequency engineering to suppress the coupling of radiation to bound surface waves supported by a metallic ground plane[18]. The key is to create a high-impedance surface that does not support bound surface modes. This is possible by patterning a metal film in such a way that the ratio of the electric over the magnetic field at the surface becomes extremely high. Such a surface was found to naturally avoid the undesirable shorting of an electric dipole antenna placed in close proximity to a high-conductivity metal film. It was also found to suppress the coupling of antenna emission to non-radiative bound surface modes. The history of high-impedance surfaces has its roots in the notion of electromagnetic hard and soft surfaces—a concept borrowed from acoustics and brought into the electromagnetic regime at microwave frequencies for RF antenna feed engineering[19–21] and for the design of artificial magnetic conducting surfaces suitable for conformal antennas[18,22]. Such surfaces were later proposed as a ground plane for thin absorbers[23,24]. With the recent appearance of several high-impedance metasurface mirror designs operating in the visible spectral range[25–30], it is logical to explore their application to enhance the emission from quantum dipole emitters located near a metallic electrode/reflector.

In this work, we provide a new solution for the longstanding fundamental challenge of using highly conductive metallic electrodes in thin film LEDs and break the usual dichotomy between high electrical and high optical performance of such electrodes. We illustrate how the detrimental excitation of SPPs and LWs can be greatly reduced by implementing a high-impedance nano-patterned metal as the electrode. We transplant the concept of high-impedance ground planes into the visible domain with the specific goal of enhancing emission from optical emitters situated on such nano-patterned metasurfaces, effectively constructing optical conformal nanoemitters and illustrate the benefits of this approach for enhancing the light emission from dye molecules placed on different types of nanopatterned metal surfaces.

## Results

### Radiative decay control with a high-impedance metasurface.
We start by showing how the radiative decay of a quantum emitter above a metal film can be modified through the introduction of a subwavelength corrugation. Figure 1a shows the electromagnetic fields generated by a dipole emitter placed in close proximity to a smooth metal surface serving as an electric mirror (EM), oscillating at a frequency corresponding to a free space wavelength of $\lambda_{em} = 560$ nm. Whereas the emitter radiates

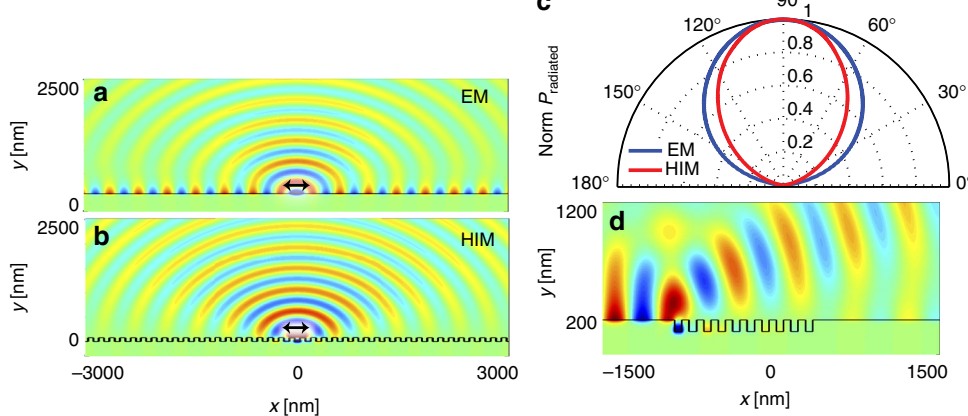

**Fig. 1** Absence of propagating SPPs on a HIM metal electrode. **a** Simulation of the magnetic field profile for an electric dipole emitting at wavelength of 560 nm and positioned 10 nm above a smooth silver film serving as an electrical mirror (EM). The orientation of the dipole is parallel to the silver surface. Whereas the dipole radiates some energy into free space, an undesired coupling to lossy waves (LWs) and surface plasmon polaritons (SPPs) is also observed. **b** Magnetic field profile for an electric dipole above a high-impedance metasurface (HIM) patterned with 100-nm-deep grooves spaced at a subwavelength periodicity of 150 nm and for a metal filling fraction of 50%. Same dipole orientation as in **a**. No emission into SPPs is observed and the radiation into free space is enhanced. **c** Broad angular emission distribution for a horizontal dipole above a smooth EM and a patterned HIM. **d** Decoupling of SPPs incident on a patterned HIM area consisting of a subwavelength array of grooves with same dimensions as in **b**. Excitation wavelength for SPPs is 560 nm

some energy into the far-field, it also excites undesired LWs and SPPs. The LWs are highly localized in the near-field of the emitter and physically capture the excitation of an image-dipole in the metal. The SPPs propagate away from the emitter along the surface and reduce in strength as they dissipate energy in the metal. After introducing a subwavelength groove-array into the metal to form a high-impedance metasurface (HIM), there is no noticeable SPP excitation on the electrode surface (Fig. 1b). The same holds true for an electric dipole that is oriented normal to the metal surface (Supplementary Fig. 2). A more detailed investigation also reveals that the HIM can effectively reduce the coupling to LWs. The creation of the grooves locally increases the emitter-to-metal spacing by removing some of the metal and thereby diminishes this loss channel (Supplementary Figs. 3, 4).

One easy way to verify whether the patterned surface serves as a HIM is to calculate the reflection phase of a normally incident plane wave. A high-impedance surface features a reflection phase between $-\pi/2$ and $\pi/2$[18], and we find a reflection phase of $0.2\pi$ for our grooved metasurface. This reflection phase can also be used to calculate the impedance[23] and this gives $3.6\eta$, where $\eta$ is the characteristic impedance of free space. This is 12 times higher than the surface impedance of the flat silver mirror. It can be seen that the far-field radiation is achieved without significantly affecting the broad angular radiation patterns of electric dipole emitters, which is desirable for many lighting and display applications (Supplementary Fig. 5).

The absence of guided SPP modes on a properly designed HIM can be verified by launching a SPP along a smooth metal film in the direction of a finite-sized HIM region. In Fig. 1d, an SPP wave is traveling toward a set of ten-subwavelength grooves is seen to effectively decouple and radiate into the far-field in a mere 1-μm-long corrugated region. Notably, little backreflection of the SPP wave occurs. Simulations show that the SPP decoupling occurs efficiently across a wide frequency range while generating little SPP reflection or transmission for groove depths ranging from 50 to 120 nm (Supplementary Fig. 6).

**Engineering a broad bandgap for surface plasmon polaritons.** The observations in Fig. 1 show that the HIM effectively transforms guided SPPs on a smooth metal surface into leaky waves. This is consistent with the calculated bandstructure of the HIM (Fig. 2a) The patterned surface exhibits a region with no allowed SPP modes in the broad frequency range from 370 to 750 THz (400 to 810 nm) due to the high impedance of the structure. The guided modes at the upper and lower edge of this region correspond to plasmonic modes that feature a high magnitude of the magnetic field at the top of the corrugations and inside the grooves. The magnetic field profile $|H_z|$ for these two modes is shown in Fig. 2b, c. At frequencies between these extremes, no guided SPP modes are allowed, and consequently SPPs incident on a HIM are either reflected or radiated into free space, with the radiation channel dominating as a result of the high impedance of the surface as shown in Supplementary Fig. 6. It is worth emphasizing that for the proposed subwavelength groove arrays (groove spacing $\lambda_{em}/4$) the formation of a bandgap is linked to the effective decoupling of quasi-guided waves from the surface. This

**Fig. 2** SPP band gap for a HIM consisting of a metal patterned with a subwavelength groove-array. **a** Dispersion relation for surface plasmon polaritons (SPPs) propagating along a smooth silver/air interface (black), a high-impedance metasurface (HIM) (red) consisting of 100-nm-deep and 75-nm-wide grooves space at period of 150 nm, and the light line in air (green). The spectral emission band of R6G molecules is also shown in the figure demonstrating that dye's emission falls into the bandgap of the high-impedance surface. **b**, **c** Spatial distribution of magnetic field magnitude $|H_z|$ for the high- and low-frequency modes of the high-impedance metasurface at the points labeled T (for top) and G (for gap) in the dispersion relation. **d** Scanning electron microscopy (SEM) and **e** optical images. The groove array is illuminated with a planar SPP wave from the top (arrows). The groove array decouples the SPPs to the far field and casts a shadow on the outcoupling groove at the bottom of the image. **f** Measured transmission spectrum for the SPPs across the groove array, defined as the ratio of light intensity scattered at the exit slit with and without the array

behavior contrasts the behavior of typical plasmonic bandgap structures that feature shallow grooves with a spacing of half a SPP wavelength for which the formation of a bandgap is linked to SPP reflection in the plane of the metal surface, rather than the enhanced radiation away from the surface achieved here.

Next, we experimentally measure the transmission of SPPs across an array of ten subwavelength grooves. The targeted groove dimensions are 100 nm deep, 75 nm wide, and the periodicity is 150 nm. To this end, an optically thick, silver film is deposited on a smooth glass substrate. Focused ion beam (FIB) milling is then used to carve a groove-array as shown in Fig. 2d. The groove-array is flanked on one side by a slit through the metal used to launch SPPs towards the groove-array. A decoupling groove is milled on the opposing side of the groove array to monitor the SPP transmission. Upon illumination of the excitation slit with a collimated white light source, both the groove array and the decoupling groove light up. The decoupling groove features a darker section directly behind the groove-array, indicating that as few as ten grooves can effectively prevent the propagation of SPP waves by converting them to far-field radiation.

The SPP transmission spectrum is shown in Fig. 2e. The transmission coefficient is less than 0.2 for wavelengths in the range from 500 to 800 nm and increases abruptly at the edges of this spectral window. The onsets of high transmission occur close to the calculated upper and lower edges of the bandgap of the patterned surface. The low transmission in the bandgap region is due to the decoupling of SPPs by the HIM across the entire visible range, rather than due to Bragg backreflection of SPPs across a narrow bandwidth, as is observed for plasmonic bandgap structures based on shallow metallic gratings[31–33](Supplementary Note 2).

**Experimental demonstrations of enhanced light emission.** We can now exploit these HIMs to increase the EQE of a light-emitting material. We cover a HIM region as shown in Fig. 3a with a 15-nm-thick layer of Rhodamine 6G (R6G) in polymethyl methacrylate (PMMA). The broad emission spectrum of R6G is peaked at 550 nm, inside the predicted band gap of the HIM discussed in Fig. 2a. In order to maintain a fixed density of emitters per unit area for different groove parameters, we first fill the grooves with $SiO_2$ to create a planar overcoat before spin-casting the dye-doped PMMA layer (see "Methods"). $SiO_2$ deposition in the grooves will slightly move band edges calculated in the Fig. 2 to 300 THz and 700 THz. This broad range still contains the spectral emission peak for R6G molecules at 530 THz. The emitter layer is then excited with a 485 nm pulsed laser. In order to minimize the effect of the excitation enhancement on fluorescence intensity, the laser is linearly polarized along the grooves (transverse electric (TE) polarization) to avoid direct excitation of gap SPPs in the grooves that could result in large absorption enhancements in the dye layer. Photoluminescence

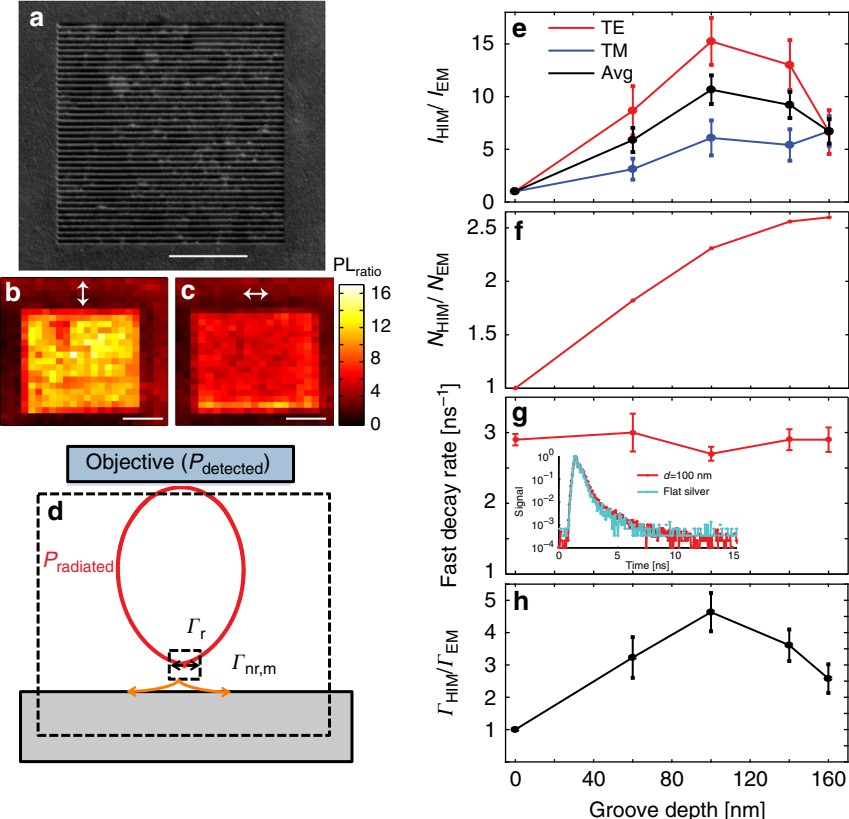

**Fig. 3** Photoluminescence intensity enhancement on a high-impedance metasurface. **a** Scanning electron microscopy (SEM) image of 100-nm-deep groove-array patterned into a smooth Ag film. Scale bar is 2 μm. **b**, **c** Photoluminescence maps of a grooved surface covered with a R6G dye layer. The excitation laser is polarized along the grooves. The detection polarization is perpendicular or parallel to the grooves, respectively. **d** Schematic of different quantities used in our analysis of the PL decay. Scale bar is 2 μm. **e** PL intensity enhancement as a function of the groove depth for TM and TE polarized collection, as well as the average. **f** Simulated enhancement of the density of excited molecules in the emissive layer as a function of groove depth. **g** Dependence of the dominant fast-decay component of the PL lifetime for molecules on top of a smooth silver film and on HIMs with different groove depth. The inset shows examples of measured lifetime traces for the case of groove depth of 100 nm and flat mirror. **h** Calculated radiative decay rate enhancement. Error bars indicate the 95% confidence interval of the fit

(PL) maps are obtained by scanning the laser excitation spot across the sample while collecting the PL emission at each point with a confocal microscope. The maps are created while collecting the PL signal over the entire emission spectrum of the dyes.

As seen in emission maps for detection polarizations oriented both parallel and perpendicular to the grooves (Fig. 3b, c), the emission intensity is relatively uniform across the patterned surface region and significantly higher than in the adjacent flat area. From these maps we obtained a PL intensity enhancement factor by averaging the PL intensity across the patterned area and dividing it by the average PL intensity from a flat sample region. The enhancement is larger for the transverse magnetic (TM) polarized emission normal to the grooves (about 15) than for the TE polarized emission (approximately 5). This is expected as the leaky waves excited by the emitting molecules are longitudinal waves with an electric field in the propagation direction (like SPPs on a smooth metal surface). This field direction is maintained upon decoupling by the HIM. The measured $I_{PL}$ enhancement depends on the groove depth for both TM and TE polarization (Fig. 3e). The ratio between TM and TE emission varies from 1 to 2.5 depending on the groove depth. It is clear that an anisotropic HIM can not only enhance emission, but also control the polarization state of the emitted light directly at the source (without the need for external components like polarizers and wave plates).

**Quantitative analysis of the photoluminescence enhancements.** To quantitatively understand the origin of this substantial PL enhancement we consider several possible contributions. Under pulsed excitation the measured PL intensity $I_{PL}$ of an ensemble of molecules is proportional to the number of excited molecules per unit area $N$ immediately after the pulse, the collection efficiency of the optics $\eta_{col}$, and the EQE $\eta_{ext}$ of the sample:

$$I_{PL} \sim N \times \eta_{col} \times \eta_{ext} \qquad (1)$$

We define the EQE as:

$$\eta_{ext} = \frac{\Gamma_r}{\Gamma_r + \Gamma_{nr,m} + \Gamma_{nr}^0} = \Gamma_r \times \tau \qquad (2)$$

where $\Gamma_r$ is the radiative decay rate for emission into the far-field either directly or after interacting with the metamirror, $\Gamma_{nr}^0$ is the intrinsic non-radiative decay rate of the molecules, and $\Gamma_{nr,m}$ is the non-radiative decay rate quantifying the dissipation of energy into the metal film. Here, we treat this rate as a non-radiative rate as none of this emission is observed in the far-field (Fig. 3d). The EQE can also be written in terms of the measured decay lifetime $\tau = (\Gamma_r + \Gamma_{nr,m} + \Gamma_{nr}^0)^{-1}$.

In light of Eqs. (1) and (2), an enhancement of emitted intensity can be attributed to an increase in the number of excited dye molecules per collection area, the collection efficiency, and the EQE due to changes in radiative and total decay rates. In the rest of this report, we look at the impact of each of these terms.

To compare the excitation of molecules on flat and patterned surfaces, we carried out full-field simulations with different groove depths (Fig. 3f). The absorption of laser light in the emissive layer as a function of groove depth reaches a maximum excitation enhancement factor of approximately 2.5. Our measurements are in the low-pump power regime (far from saturation). In this regime, the number of excited molecules is linearly proportional to the absorption enhancement.

Regarding the collection efficiency, we can assume that $\eta_{col}$ is essentially the same for patterned and unpatterned areas for two reasons. First, we use a high numerical aperture objective (NA =

0.95). Second, our simulations indicate that the angular radiation pattern of a dipole above the patterned HIM is virtually unchanged compared to a smooth metal surface (see Fig. 1c and Supplementary Figs. 2, 5).

To understand the changes in EQE, which depends on the radiative and total decay rates, we measure PL decay traces (Fig. 3g and Supplementary Fig. 7). The traces for the HIM and flat silver mirror both show a largely single-exponential decay, indicating that the PL signal predominantly originates from molecules with a single, fast decay rate. Despite the large observed PL enhancement for some groove depths, the decay rate is found to vary by less than 10% for different groove depths. We attribute the small change in lifetime of the emitters to a relatively high intrinsic non-radiative decay rate of R6G molecules when embedded in our chosen polymer matrix. Indeed, the same molecules on a glass substrate have a fast rate of 2.5 ns$^{-1}$, close to 2.9 ns$^{-1}$ measured on an unpatterned silver mirror.

It is thus clear that the small changes in the excitation efficiency (about 2.5) and insignificant changes in the collection efficiency and PL decay rate cannot explain the large changes in the PL intensity (approximately 15). These changes can therefore largely be attributed to an enhancement in the radiative rate, for example a redirection of the emission into the far-field instead of into the metal as expected for a high-impedance surface that suppresses SPPs and LWs. It should be noted that unlike photonic bandgap structures in dielectrics where the local density of optical states goes to zero inside the bandgap, HIMs trade bound surface modes with leaky modes. As a result, an emitter can couple its emission to these leaky modes, which eventually contribute to the farfield radiation. The enhancement in the radiative rate can be assessed by dividing the measured PL intensity enhancement by the simulated absorption enhancement (Fig. 3h): a maximum enhancement of the radiative decay rate of approximately five can be reached with a groove depth of 100 nm. As the relative fractions of the power coupled toSPPs versus free space on a smooth metal just depends on thedistance of the emitter to the metal and not on the internal quantumefficiency, this approach is effective for both low and high internal-quantum-efficiency emitters.

Regarding the electrical performance of such electrodes, given the fact that the resistivity of metals like Ag and Al is two or three orders of magnitude lower than a good transparent conductive oxides (TCOs), the use of patterned continuous metal films can outperform continuous TCOs in terms of their charge injection properties. (Supplementary Note 3).

A periodic array of subwavelength grooves can thus increase the PL intensity by increasing the EQE of the emissive layer, while controlling the polarization state of the emission. However, a polarization-independent enhancement in the PL intensity is also possible. To this end, we fabricated high-impedance surfaces based on dimples filled with $SiO_2$ arranged on a square lattice (Fig. 4). Band structure calculations of such a polarization-independent high-impedance surface predict a clear band gap that spans a broad wavelength range 485–600 nm (Supplementary Fig. 8). We also calculated the reflection phase of such a metasurface for normal incident light at visible frequencies (Supplementary Fig. 8b). We find values that are consistent with observations by others at RF frequencies[18]. We used the same fabrication process as for the linear grooves to fill the dimples with $SiO_2$ and deposit an emissive layer on top of the high-impedance surface. Again, PL maps demonstrate PL enhancement, but in this case the enhancement is independent of the detection polarization. The enhancement increases with dimple depth and reaches a maximum value of approximately 12 for a groove depth of 180 nm (Fig. 4d), similar to the maximum TM polarization value for grooved samples but now obtained for both

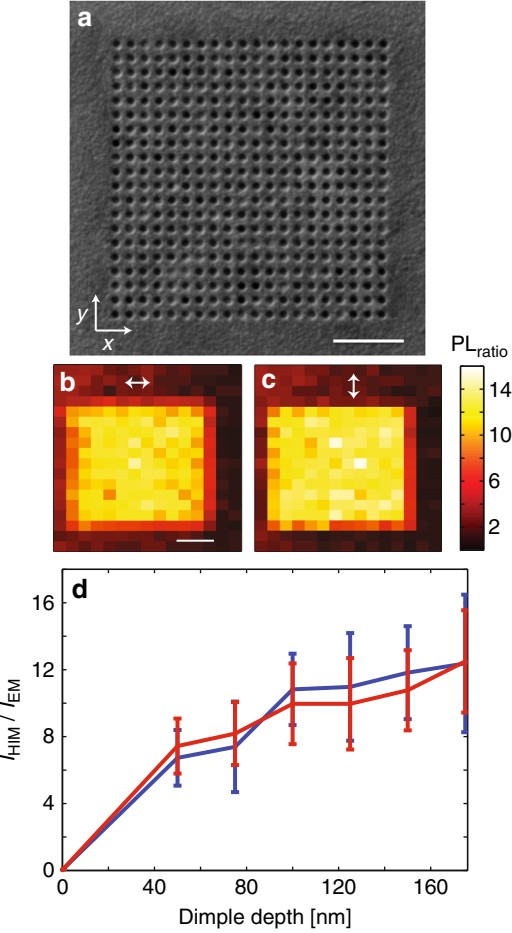

**Fig. 4** Polarization-independent photoluminescence enhancement achieved with a high-impedance metasurface featuring an array of nanodimples. **a** Scanning electron microscopy (SEM) image of a subwavelength dimple array patterned into a silver substrate. Scale bar is 1 μm. **b**, **c** PL maps of molecules placed on a HIM with nanodimples and for two orthogonal polarization directions for the optical collection and a fixed excitation polarization along the x-axis. Scale bar is 1 μm. **d** Measured PL intensity enhancement for polarizations along the x-axis (blue) and y-axis (red) as a function of dimple depth. Error bars show the standard deviation of enhancement across patterned area

polarizations. Similar to the linear groove arrays, the lifetime measurements do not show any significant decay rate differences for the molecules on the dimple arrays compared to those on a flat sample region. We conclude that similar physics underlies the observed PL enhancement for both dimple and groove arrays.

## Discussion

Motivated and inspired by the notion of high-impedance ground planes in the radio frequency regime, we have developed an optical version of this concept for enhancing optical emission. We demonstrated the realization of high electrical conductivity metallic electrodes that combine high reflectivity at visible frequencies with significantly reduced optical losses compared to traditional metallic electrodes. These HIMs can enhance the light extraction efficiency of quantum emitters and reduce undesired optical losses in metallic electrodes by introducing a wide SPP band gap and mitigating dissipation in LWs. Within the gap no guided surface waves exist and the emission is rapidly decoupled

to free-space. This approach is effective for both low and high internal-quantum-efficiency emitters. The broadband nature of the SPP band gap observed for these metasurfaces makes them promising candidates to improve the performance of solid-state light-emitting devices for illumination and display applications. Moreover, these metasurfaces provide an excellent platform for engineering and manipulating SPPs along two-dimensional structures (flat photonics), allowing additional functionality in light-matter interaction.

## Methods

**Device fabrication**. HIMs were fabricated in a multistep process. Silicon wafers with 300-nm-thick thermally grown oxide were used as a planar and smooth substrate. They were cleaned in several steps. First, they were sonicated with Alconox precision cleaner solution, then rinsed with water three times. After this step, they were sonicated for 5 min in Acetone, Methanol, and Isopropanol, consecutively, and finally dried with nitrogen gas. A smooth, optically thick (300 nm) silver film was then deposited onto the substrate using a 2 nm germanium nucleation layer. The metal was nanopatterned by FIB milling with a FEI Helios 600i dual FIB/SEM tool. A thick layer of silicon oxide was deposited on the patterned area with electron beam assisted deposition for the first 50 nm and ion beam assisted deposition for the next 250 nm. Then the deposited $SiO_2$ layer was milled down to the silver surface with FIB until a 5~10 nm overcoat of $SiO_2$ was left on the high-impedance area and the surrounding flat area.

To deposit a thin light-emitting layer, Rhodamine 6G powder was dissolved in a Polymethyl methacrylate (PMMA) solution. The PMMA was diluted with Anisole to reduce the achievable thickness of the emissive layer. The solution was then put in a vortex mixer for 2 min. The concentration of R6G in the PMMA solution is approximately $3 \times 10^{-4}$ M. The thickness of the R6G film is around 15 nm, as measured by SEM imaging the cross section of our samples.

**Optical measurements**. PL maps were acquired with a Witec alpha 300R confocal microscope with a two-axis piezo stage for sample scanning. A PicoQuant LDH-P-C-485 pulsed laser at wavelength of 485 nm is used for excitation of R6G molecules. The PL signal is detected with an avalanche photodiode (MPD APD) for both lifetime measurement and PL intensity measurement. The spatial resolution of the imaging system in our experiments is about 500 nm. The measured lifetime traces were fitted with a bi-exponential function of the form $I(t) = ae^{-r_1 t} + (1-a)e^{-r_2 t}$ where $a$ is the amplitude of the fast contribution and $r_1$ and $r_2$ are the decay rates of the two contributions. The value of $a$ is larger than 0.99 for all five groove depths (Supplementary Fig. 6)

**Simulations**. All simulations were carried out using the Finite-Difference Time-Domain method (Lumerical FDTD Solutions).

**Data availability**. All data are available from the authors on reasonable request.

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

## Acknowledgements

This work was supported by a Multi University Research Initiative (MURI FA9550-12-1-0488) from the AFOSR and a gift from Konica Minolta Laboratory USA. A.G.C. also acknowledges the support of a Marie Curie International Outgoing Fellowship.

## Author contributions

M.E. performed all simulations and modeling and fabricated the samples in this study. M.E. and A.G.C. performed the optical experiments. M.E., A.G.C., P.G.K., N.E., and M.L.B. were involved in the physical analysis of the measurement results. M.L.B. supervised the project.

## Additional information

**Competing interests:** The authors declare no competing interests.

