## [Peer Review File · Nature Communications]

Reviewers' comments:

Reviewer #1 (Remarks to the Author):

In the manuscript, entitled "Optical emission near a high-impedance mirror," by M. Esfandyarpour, et al., the authors introduce and study, both experimentally and computationally, an optical 'high electromagnetic impedance' meta-surface (engineered surface) specifically for enhancing optical emission from an emitter (Rhodamine 6G). They carefully study the physics behind this effect, which they show is essentially owing to the enhanced coupling of the emitted radiation to free space rather than to surface plasmon polaritons that are eventually absorbed in the metal, and they finally (Fig. 4) demonstrate a polarization-independent photoluminescence enhancement in a judiciously designed 2D metasurface. The reported maximum enhancement of the radiative decay rate is of the order of 5.

I found this work to be clearly/nicely written and fairly novel (and of potential interest in the design of improved light-emitting devices and displays), in principle suitable for publication in a journal such as NCOMMs. Below, I only have a few comments, the addressing of which, I feel, could further enhance the clarity/quality of the work:

- 1) Throughout the manuscript the authors talk about (and justify the terminology use of) 'high-impedance' metasurface, but there is no mention of the actual/quantified value of the high impedance(s) of the used metasurface(s). How large are the impedances of the herein-designed metasurfaces, and how do they e.g. compare with that of free space?
- 2) For electrical pumping, will the fact that the new/structured metallic electrodes have 'holes'/grooves in them (less metallic surface at the very top of the metallic electrodes) lead to fewer carriers being able to be injected in the active region?
- 3) Does the addition of SiO₂ in the grooves of the metasurface(s) modify their optical properties (e.g., the coupling of SPPs to free space)?
- 4) In the inset of Fig. 3(g), why are the two rates almost the same? (in fact, the one for the flat-silver case somewhat appears to even be very slightly faster...)
- 5) Where are the error bars in Fig. 3(h)? (cf. caption of that figure part)
- 6) It is not currently very clear why, in Fig. S2(a), the loss ratio for the metasurface for a dipole distance of around 80 nm is higher than that of a flat Ag mirror. Is that, e.g., a numerical artifact of the particular way in which losses were calculated in that result? - please explain in a bit more detail.
- 7) Some relevant, recent references (on nanoantenna-aided LEDs, active nanophotonics, etc) are currently missing from the manuscript [e.g., Nature Materials 11, 573 (2012), Nature Photonics 9, 427 (2015), Optics Express 24, 17916 (2016), etc], and could be included for the sake of completeness and for the prospective readers of this work.
- 8) p. 3, below the middle of first paragraph: a typo ("fromf") should be amended.

Reviewer #2 (Remarks to the Author):

In this paper, the authors design 1D and 2D high impedance surfaces at visible wavelengths by milling subwavelength-spaced linear grooves and a 2D array of subwavelength-spaced dimples, respectively,

into a metallic surface.

A quantum light emitting material (Rhodamine 6G) is placed atop the high impedance surface, and significant enhancement in photoluminescence is observed over the case where it is placed atop a flat metallic surface. The authors claim that the photoluminescence is due to an increase in radiative decay rate caused by the presence of the surface. The reported high impedance surfaces could have a major impact in the solid state display and lighting industries, given that they can both act as metallic electrodes but also prevent the propagation of surface plasmon polaritons that trap and dissipate light.

The paper is interesting and of high quality. However, the following questions need to be addressed before it can be further considered for publication in Nature Communications.

- 1) The authors should state the DC electrical conductivity of the grooved/dimpled surfaces?
- 2) Does Figure 1D show diffraction from the edge of the high impedance surface, or does it show leaky-wave radiation? The authors claim that the excitation of leaky waves is substantiated by the HIM band structure shown in Fig. 2a. However, I have doubts. Leaky-wave excitation requires a propagation band (with complex propagation constant) within the light cone. The band diagram in Fig. 2a only shows bands outside the light cone. In addition, if there was a leaky-wave excitation, the emitted pattern would be a fan beam, not the dipole pattern shown in figure 1c. Further, the radiation pattern should scan with frequency. I do not doubt the enhancement in photoluminescence reported by the authors, I just do not think the leaky-wave explanation given for it is correct.
- 3) It seems that the high impedance surface was initially designed assuming the grooves were air filled, and then in the fabrication stage the grooves were filled with SiO₂. How much did the resonant frequency (high impedance condition) shift (in terms of wavelength) once the grooves were filled. Also how much did the stopband bandwidth reduce by?
- 4) Why is the maximum excitation enhancement factor only ~2.5 between the patterned and flat surfaces? It seems like it should be higher given that the two scenarios represent a dipole (dye molecule) on an electric conductor and a dipole on a magnetic conductor?

Reviewer #3 (Remarks to the Author):

The authors analyze whether light extraction efficiency of quantum emitters can be enhanced by placing them close to a metal surface, patterned such that it behaves as a high-impedance meta-surface. The rationale is that such meta-surfaces do not support surface plasmons which, in some circumstances, are a strong decay channel.

Although the idea may be interesting, I find the manuscript very poor in detail, specially concerning the role of electromagnetic non-radiative modes which are responsible for the quenching of optical emitters close to metal surfaces. The authors state that "it would be highly desirable to tackle the dissipative loss in the metal at its root and remove the coupling of emitters to SPPs on the metallic contact without the need for spacer layers and complex light-extraction schemes", because "the use of such spacers increases the device's contact resistance and the probability of trapping light inside the high-index light emitting materials via total internal reflection". But usually, spacers between the emitters and the metal surface (with a thickness of around 20nm) are placed in order to avoid quenching of emission into non-radiative non-plasmonic modes.

So, in the simulations reported in Fig. 1, the authors should compute which fraction of the emission goes into radiation and how much goes into heating the metal. In other words, what the Purcell factor of the emitter in the structure is, and out of this, how much goes into radiation. This is important because, perhaps, the structure they propose can transfer to radiation modes the energy that would otherwise couple to plasmons, but this could be a very small portion of the total energy that leaves the emitter.

This also applies to the experiment. If I understand properly, in the considered structure the emitters are at a distance smaller than 15nm than the metal (at least the emitters that do not sit inside the groove opening). Can the authors provide an absolute figure on the emitted power, in terms of the incident power? can they estimate how much energy goes into heating the metal?.

All this characterization is, in my opinion, needed, at least to discard (or estimate the relevance) of another mechanism that may be present in their structure: by creating grooves part of the emitters are placed further away from the metal, and thus quenching may be reduced.

This possible alternative interpretation could also explain why the proposed structure works both for grooves and dimples. despite the fact that the electromagnetic response of grooves and dimples can be very different, as the field propagation inside them is very different. In fact I am puzzled by the result for dimples: in order to have a high-impedance surface it is my understanding that the field must penetrate substantially into the hole, but most probably the field decays exponentially inside the hole for the parameters considered. I believe that the authors should demonstrate that the considered dimple-metasurface is really a high impedance one, as they claim.

To summarize, in my opinion the manuscript presents results that may be potentially interesting for a specialist, but the analysis is rather poor. I do not recommend publication, even less in a high-quality journal as Nat. Comm.

Response to the Referees

Reviewer #1 (Remarks to the Author):

General comments from Reviewer 1

In the manuscript, entitled "Optical emission near a high-impedance mirror," by M. Esfandyarpour, et al., the authors introduce and study, both experimentally and computationally, an optical 'high electromagnetic impedance' meta-surface (engineered surface) specifically for enhancing optical emission from an emitter (Rhodamine 6G). They carefully study the physics behind this effect, which they show is essentially owing to the enhanced coupling of the emitted radiation to free space rather than to surface plasmon polaritons that are eventually absorbed in the metal, and they finally (Fig. 4) demonstrate a polarization-independent photoluminescence enhancement in a judiciously designed 2D metasurface. The reported maximum enhancement of the radiative decay rate is of the order of 5.

I found this work to be clearly/nicely written and fairly novel (and of potential interest in the design of improved light-emitting devices and displays), in principle suitable for publication in a journal such as NCOMMs. Below, I only have a few comments, the addressing of which, I feel, could further enhance the clarity/quality of the work:

Response to the general comments from Reviewer 1

We would like to thank this reviewer for his/her positive evaluation of the science and its presentation.

Comment 1 from Reviewer 1

Throughout the manuscript the authors talk about (and justify the terminology use of) 'high-impedance' metasurface, but there is no mention of the actual/quantified value of the high impedance(s) of the used metasurface(s). How large are the impedances of the herein-designed metasurfaces, and how do they e.g. compare with that of free space?

Response to comment 1 from Reviewer 1

One easy way of obtaining the surface impedance of a patterned surface is to calculate the reflection phase of a normally incident plane wave from the surface. Borrowing from RF frequency studies, a high impedance surface features a reflection phase between $-\pi/2$ and $\pi/2$. At a selected wavelength of interest at 560 nm, we find a reflection phase of 0.2π for our grooved metasurface. The reflection phase can also be linked to the impedance as shown e.g. in reference 23. The impedance corresponding to 0.2π is 3.6η where η is the characteristic impedance of free space, which is 12 times higher than the surface impedance of a flat silver mirror. To achieve a high impedance, the depths of the grooves should be chosen to be roughly $1/4$ of the wavelength of the gap surface plasmon in the grooves. We now mentioned these points in the text.

Comment 2 from Reviewer 1

For electrical pumping, will the fact that the new/structured metallic electrodes have 'holes'/grooves in them (less metallic surface at the very top of the metallic electrodes) lead to fewer carriers being able to be injected in the active region?

Response to comment 2 from Reviewer 1

If a good electrical contact can be realized between the metallic metasurface and the emitting semiconductor layers than this should not be an issue. The argument for this comes from a basic comparison of the electrical resistance of the designed high impedance metasurface with TCO

spacer layers. We calculated the resistance of a periodic array of metallic nanobeams that make up the high-impedance metasurface. Such an array can be treated as a set of parallel resistors. In this calculation we consider the beam-array with periodicity P , a silver beam width of a , and height h . A small filling fraction of metal $f = a/P$ increases the resistance of a MM mirror that is given by $R_{HIM} = \rho (h/Af)$, where ρ is the resistivity of silver and A is the area of the HIM mirror. Similarly, the resistance of a TCO layer of an equivalent thickness h can be written as $R_{TCO} = \rho (h/A)$. The magnitude of f of practical MM mirror is between 0.1-0.9 while the resistivity of metals like Ag and Al is two or three orders of magnitude lower than a good TCO. As a result, the electrical conductivity of a HIM mirror will always be higher than those of a TCO spacer. We now mention this in the main text.

Comment 3 from Reviewer 1

Does the addition of SiO₂ in the grooves of the metasurface(s) modify their optical properties (e.g., the coupling of SPPs to free space)?

Response to comment 3 from Reviewer 1

The addition of SiO₂ in the grooves will modify the optical properties, as the effective mode index of the gap plasmon mode supported by the grooves will change. The band structure calculation shown in figure 2 are done considering grooves filled with air because the transmission measurement shown in figure 2 are done with a sample that has no SiO₂ deposited in the grooves. If we do the same band structure calculation for a device that has grooves filled with higher-index silica, we find that the band edges will move slightly to 300 THz and 700 THz. This broad range still nicely contains the spectral emission peak for R6G molecules at 530 THz. We have added one sentence in the main text explaining the effect of SiO₂ on the optical response of the electrode to make this important point clear.

Comment 4 from Reviewer 1

In the inset of Fig. 3(g), why are the two rates almost the same? (in fact, the one for the flat-silver case somewhat appears to even be very slightly faster...)

Response to comment 4 from Reviewer 1

In the main text we mention that the small change in lifetime of the emitters can be attributed to a relatively high intrinsic non-radiative decay rate of R6G molecules when embedded in our chosen polymer matrix. Indeed, the same molecules embedded in this polymer matrix on a glass substrate have a fast rate of 2.5 ns⁻¹, close to 2.9 ns⁻¹ measured on an unpatterned silver mirror.

Comment 5 from Reviewer 1

Where are the error bars in Fig. 3(h)? (cf. caption of that figure part)

Response to comment 5 from Reviewer 1

We would like to thank the reviewer for noting the missing error bars. We have added them in the revised version.

Comment 6 from Reviewer 1

It is not currently very clear why, in Fig. S2(a), the loss ratio for the metasurface for a dipole distance of around 80 nm is higher than that of a flat Ag mirror. Is that, e.g., a numerical artifact of the particular way in which losses were calculated in that result? - please explain in a bit more detail.

Response to comment 6 from Reviewer 1

The loss ratio above any reflecting surface features wavelength-scale oscillations. This is expected as the radiation from a dipole emitter can follow two distinct pathways that can interfere. These are the pathways directly to free space or after a reflection from the mirror. For grooved metamaterial mirrors, the reflection phase is shifted and the maximums and minimums in the loss ratio occur at a different distance from the flat Ag mirror. We now mention this point in the caption to Fig. S2.

Comment 7 from Reviewer 1

Some relevant, recent references (on nanoantenna-aided LEDs, active nanophotonics, etc) are currently missing from the manuscript [e.g., Nature Materials 11, 573 (2012), Nature Photonics 9, 427 (2015), Optics Express 24, 17916 (2016), etc], and could be included for the sake of completeness and for the prospective readers of this work.

Response to comment 7 from Reviewer 1

These are excellent references and we have added them to the manuscript.

Comment 8 from Reviewer 1

p. 3, below the middle of first paragraph: a typo ("fromf") should be amended.

Response to comment 8 from Reviewer 1

We took care of this typo.

Reviewer #2 (Remarks to the Author):

General comments from Reviewer 2

In this paper, the authors design 1D and 2D high impedance surfaces at visible wavelengths by milling subwavelength-spaced linear grooves and a 2D array of subwavelength-spaced dimples, respectively, into a metallic surface.

A quantum light emitting material (Rhodamine 6G) is placed atop the high impedance surface, and significant enhancement in photoluminescence is observed over the case where it is placed atop a flat metallic surface. The authors claim that the photoluminescence is due to an increase in radiative decay rate caused by the presence of the surface. The reported high impedance surfaces could have a major impact in the solid state display and lighting industries, given that they can both act as metallic electrodes but also prevent the propagation of surface plasmon polaritons that trap and dissipate light.

The paper is interesting and of high quality. However, the following questions need to be addressed before it can be further considered for publication in Nature Communications.

Response to general comments from Reviewer 2

We thank this reviewer for the overall very positive evaluation of our paper.

Comment 1 from Reviewer 2

The authors should state the DC electrical conductivity of the grooved/dimpled surfaces?

Response to comment 1 from Reviewer 2

For a basic comparison of the electrical resistance of designed high impedance metasurface with TCO spacer layers, we calculated the resistance of a periodic array of metallic nanobeams that make up the High impedance metasurface. Such an array can be treated as a set of parallel resistors. In this calculation we consider the beam-array with periodicity P , a silver beam width of a , and the thickness of the layer is h . A small filling fraction of metal $f = a/P$ increases the resistance of the high impedance metasurface is given by $R_{\text{HIM}} = \rho (h/Af)$, where ρ is the resistivity of silver and A is the area of the HIM mirror. Similarly, the resistance of a TCO layer can be written as $R_{\text{TCO}} = \rho (h/A)$. The magnitude of f of practical MM mirror is between 0.1-0.9 while the resistivity of metals like Ag and Al is two or three orders of magnitude higher than a high quality TCO material. As a result, the electrical conductivity of a HIM mirror will always be higher than those of a TCO spacer.

Comment 2 from Reviewer 2

Does Figure 1D show diffraction from the edge of the high impedance surface, or does it show leaky-wave radiation? The authors claim that the excitation of leaky waves is substantiated by the HIM band structure shown in Fig. 2a. However, I have doubts. Leaky-wave excitation requires a propagation band (with complex propagation constant) within the light cone. The band diagram in Fig. 2a only shows bands outside the light cone. In addition, if there was a leaky-wave excitation, the emitted pattern would be a fan beam, not the dipole pattern shown in figure 1c. Further, the radiation pattern should scan with frequency. I do not doubt the enhancement in photoluminescence reported by the authors, I just do not think the leaky-wave explanation given for it is correct.

Response to comment 2 from Reviewer 2

There is indeed a propagation band within the light cone, as verified from our bandstructure simulation. They are much harder to see/detect in such simulations as the propagation along the surface are very short range due to the effective decoupling from the surface. The presence of the proposed leaky modes is more easy to see in simulations such as shown in Figure 1d. This simulation shows how the grooved metasurface converts a surface plasmon wave with a well-defined propagation constant and incident from the left into a leaky surface wave with a propagation constant that is small enough to lie within the light cone. The presence of a small propagation constant (i.e. above the light line) leaky wave can be seen from the fields inside the grooves that display a slow phase progression along the surface. The leaky surface wave decouples into a fan beam that decouples from the surface, as expected by the reviewer. The radiation pattern from a point dipole, which can excite a broad spectrum of in-plane k-vectors, resembles a dipolar pattern (See Figure 1c). Figure 1c that approximates that of. Note that this is consistent with the behavior of high impedance surfaces in the RF and THz regimes.

Comment 3 from Reviewer 2

It seems that the high impedance surface was initially designed assuming the grooves were air filled, and then in the fabrication stage the grooves were filled with SiO₂. How much did the resonant frequency (high impedance condition) shift (in terms of wavelength) once the grooves were filled. Also how much did the stopband bandwidth reduce by?

Response to comment 3 from Reviewer 2

The SiO₂ was added to appropriately space the emitter molecules from the metal surface. The addition of SiO₂ in the grooves will indeed modify the optical properties, as the effective mode index of gap plasmon mode in the grooves will change. The band structure calculation shown in figure 2 was performed considering the grooves filled with air because the transmission measurement shown in figure 2 was done with a sample that has no SiO₂ deposited in the grooves. If we do the same band structure calculation for a device that has grooves filled with a material of index 1.4 the band edges will move slightly to 300THz and 700 THz while the emission of R6G molecules is peaked at 530 THz. We now mention this point in the text.

Comment 4 from Reviewer 2

Why is the maximum excitation enhancement factor only ~2.5 between the patterned and flat surfaces? It seems like it should be higher given that the two scenarios represent a dipole (dye molecule) on an electric conductor and a dipole on a magnetic conductor?

Response to comment 4 from Reviewer 2

In our experiments, the excitation laser is polarized parallel to the direction of the fabricated grooves. For this polarization the incident light does not couple to the gap plasmon modes in the grooves. As a result, the reflection is similar to the case of the smooth metal surface with corresponding low electric fields and low excitation efficiency, and the impact on the reflection phase is minimal. This Reviewer may have been under the impression that we were exciting with light polarized orthogonal to the grooves. In that case there would have been a much bigger enhancement factor.

Reviewer #3 (Remarks to the Author):

General comments from Reviewer 3

The authors analyze whether light extraction efficiency of quantum emitters can be enhanced by placing them close to a metal surface, patterned such that it behaves as a high-impedance meta-surface. The rationale is that such meta-surfaces do not support surface plasmons, which in some circumstances, are a strong decay channel.

Although the idea may be interesting, I find the manuscript very poor in detail, specially concerning the role of electromagnetic non-radiative modes which are responsible for the quenching of optical emitters close to metal surfaces. The authors state that “it would be highly desirable to tackle the dissipative loss in the metal at its root and remove the coupling of emitters to SPPs on the metallic contact without the need for spacer layers and complex light-extraction schemes”, because “the use of such spacers increases the device’s contact resistance and the probability of trapping light inside the high-index light emitting materials via total internal reflection”. But usually, spacers between the emitters and the metal surface (with a thickness of around 20nm) are placed in order to avoid quenching of emission into non-radiative non-plasmonic modes.

So, in the simulations reported in Fig. 1, the authors should compute which fraction of the emission goes into radiation and how much goes into heating the metal. In other words, what the Purcell factor of the emitter in the structure is, and out of this, how much goes into radiation. This is important because, perhaps, the structure they propose can transfer to radiation modes the energy that would otherwise couple to plasmons, but this could be a very small portion of the total energy that leaves the emitter.

This also applies to the experiment. If I understand properly, in the considered structure the emitters are at a distance smaller than 15nm than the metal (at least the emitters that do not sit inside the groove opening). Can the authors provide an absolute figure on the emitted power, in terms of the incident power? can they estimate how much energy goes into heating the metal?.

All this characterization is, in my opinion, needed, at least to discard (or estimate the relevance) of another mechanism that may be present in their structure: by creating grooves part of the emitters are placed further away from the metal, and thus quenching may be reduced.

This possible alternative interpretation could also explain why the proposed structure works both for grooves and dimples. despite the fact that the electromagnetic response of grooves and dimples can be very different, as the field propagation inside them is very different. In fact I am puzzled by the result for dimples: in order to have a high-impedance surface it is my understanding that the field must penetrate substantially into the hole, but most probably the field decays exponentially inside the hole for the parameters considered. I believe that the authors should demonstrate that the considered dimple-metasurface is really a high impedance one, as they claim.

To summarize, in my opinion the manuscript presents results that may be potentially interesting for a specialist, but the analysis is rather poor. I do not recommend publication, even less in a high-quality journal as Nat. Comm.

Response to general comments from Reviewer 3

We are glad to hear that the Reviewer thinks that work is interesting. That said, this reviewer also brings several important concerns about the possible importance of non-radiative non-plasmonic

modes in the quenching of light emission from emitters. The non-radiative, non-plasmonic modes, also known as Lossy Waves (LWs) in the literature, are highly-localized, high-spatial-frequency modes that physically capture the excitation of image-dipoles of the emitter-dipoles in the metal. The excitation of the LWs results in light to heat conversion in a region that is very local, in the near-field of the emitter. Based on their distinct spatial frequencies, the relative excitation efficiencies of SPPs and LWs can be quantified (e.g. Chance, Prock, Silbey, Barnes, Ford and Weeber).

The reviewer correctly notes that dissipation in the metal can occur due to the coupling to LWs and SPPs on smooth metal surfaces and our proposed electrodes. In first version of the manuscript we demonstrated the beneficial impact of a subwavelength patterning of a metal electrode on the emitted light intensity. The beneficial impacts were linked to the reduction of optical losses in the metal, but we did not separately address how the patterning affects SPPs and LWs. Instead, we only argued that the patterning was beneficial in decoupling the SPPs. We agree with this reviewer that a deeper understanding of both loss processes would be desirable. This could provide further insights into the operation of our electrodes and how we can optimize them further. Experimental measurements that provide accurate, quantitative information of the magnitude and spatial distribution of the energy deposition in the metal from single/few emitters would be very interesting, but also extremely challenging. To explore the importance of LWs and SPPs, we have performed additional simulations and these have led to a greater, more nuanced insight into the beneficial role of creating a high-impedance metasurfaces.

To understand the contributions of the coupling to SPPs and LWs to the total quenching of the molecules, we performed full-field simulations of an electric dipole in 4 different scenarios. We start with the well-studied problem of an emitter above a smooth metal film. Then we analyze the impact of digging a groove underneath the emitter, and finally we show the impact of the groove-array.

The figure below shows a simulation of an electric dipole emitter radiating at 560 nm and placed 10 nm above a flat Ag mirror. The orientation of this dipole is along the x-direction, i.e. parallel to the metal surface. The simulation in Figure panel (a) clearly shows the excitation of SPPs propagating away from the emitter along the metal surface. The inset shows a zoom-in of the region near the emitter and highlights the excitation of high-spatial-frequency lossy waves. From this simulation, we calculate a high loss ratio of 70%, defined as the power going through the monitor below the dipole to the total radiated power by the dipole. From a spatial frequency analysis of the fields, we find that 60% of the loss in the metal is attributable to SPP excitation and the remaining 40% is associated with the excitation of LWs.

Second, we analyze the emission for the same dipole emitter, but now placed above a single, 75-nm-wide groove carved into the metal (Panel b). In this case, the excitation of LWs is highly suppressed as the metal is removed directly underneath the emitter. As a result, the excitation of SPPs will constitute the dominant mechanism by which energy is dissipated into the metal. With the removal of one loss channel, one might expect the fraction of optical energy deposited in the metal to reduce significantly. However, the simulation indicates that the loss ratio is virtually unchanged (from 70% to 68%). This is explained by an increase in the SPP loss that compensates for the losses to LWs. The physical mechanism for the enhanced SPP excitation lies in the effective coupling of the emitter dipole to gap SPPs that resonate in the (approximately quarter gap-SPP wavelength deep) groove, followed by an effective coupling of the gap SPPs to SPP on the metal surface. The effective excitation of the gap SPPs is visible in the inset to panel b.

Third, we analyzed the emission of a dipole in the center of the air-gap above a periodic array of nano-grooves (panel c). This geometry shows the important role of the periodic groove-arrays that have been added on either side of the central groove with the emitter. It can be seen that the SPPs on the surface are effectively decoupled and loss ratio has now dropped to 26%. This shows the importance of creating a high-impedance metasurface (i.e. a properly designed groove-array) for decoupling the SPPs.

It is also important to look at emitters in other high-symmetry locations to achieve a more complete picture. Figure panel (d) shows a simulation of the emission the same emitter-dipole, but now placed just above the center of the metal beam of the metasurface. Here, the local environment to the emitter is hardly changed and the coupling to LWs is similar to that on a smooth metal film. The currently-designed, high-impedance metasurface does not make it possible to remove this loss component, although it may ultimately be possible to produce high-impedance metasurfaces with thinner teeth. However, the neighboring grooves again do help to decouple the SPPs from the surface and the loss ratio compared to the smooth surface is reduced 34%. The results for emitters with different locations and orientations differ quantitatively, but are qualitatively the same; Grooves underneath the emitter reduce the excitation of LWs and the groove-arrays significantly aid in the decoupling SPPs. We added a section in the supplementary material of the paper to discuss the effect of lossy waves.

The dimples have a similar impact and our bandstructure calculations presented in Figure S7 indicate that the dimpled surfaces can indeed also serve as a high-impedance metasurface. As per the Reviewer's thinking, it is harder for the field to deeply penetrate into a circular hole than a groove. For this reason, the dimpled high-impedance metasurfaces feature holes with diameters of 130 nm, much larger than the 75 nm width of the linear grooves in the grooved-metasurface.

Reviewers' comments:

Reviewer #1 (Remarks to the Author):

In their replies, but also in the revised version of their manuscript, the authors have indeed adequately addressed all the points I previously brought to their attention. They are providing sufficient clarifications to justify their main claims and to help prospective readers reproduce their simulation and experimental results - and the revised version of this work looks substantially improved to me. Therefore, from my perspective, I feel that this amended work is now suitable to be published as is in NCOMMs.

Reviewer #2 (Remarks to the Author):

The authors have addressed my comments/concerns except for one point in comment 2.

In their response to comment 2, the authors write, "There is indeed a propagation band within the light cone, as verified from our bandstructure simulation."

Where is this propagation band within the light cone shown? I do not see it in Fig. 2a. Please add a sentence clarifying this point in the manuscript.

One more minor comment, the authors may want to clarify which harmonic radiates from the grooves. I believe it is the $n=-1$ harmonic rather than the fundamental.

Reviewer #3 (Remarks to the Author):

In my previous report I put forward two arguments that, in my opinion, needed to be addressed in order to substantiate the main claims of the manuscript.

The first one was related to the elimination of the need of spacer layers, as the proposed structure avoids the coupling to SPPs. In their answer the authors provide calculations for the behavior of dipoles placed at 10nm from the surface. But this calculation can only, at best, imply that the spacer layer should be of the order of 10nm. Clearly dipoles closer to the metal surface than this will present a larger coupling to lossy waves, thus being most probably quenched. So, I do not find the author's reply conclusive.

The second comment was about surprise that dimples of this size can also be used to create a high-impedance mirror. This is due not only to the smaller electromagnetic cross-section of dimples, but also on the EM mode structure inside them. If the metal were a perfect electrical conductor, holes with diameters of 130nm would not support propagating modes at wavelengths of 560nm, so their behavior should be very different to that of slits. It could be that the finite dielectric constant of silver would create a propagating mode, but I doubt it, as the induced change in propagation constant should be huge. In any case, to probe that the proposed metasurface behaves as the authors claim, the authors should have provided a study of the electromagnetic modes, and the associated effective impedances.

So, in my opinion, the authors have not addressed my comments in a satisfactory way.

Therefore, I maintain my recommendation of not publishing this manuscript in a high-quality journal as Nat. Comm.

Response to the Reviewers

Reviewer #1 (Remarks to the Author):

General comments from Reviewer 1

In their replies, but also in the revised version of their manuscript, the authors have indeed adequately addressed all the points I previously brought to their attention. They are providing sufficient clarifications to justify their main claims and to help prospective readers reproduce their simulation and experimental results - and the revised version of this work looks substantially improved to me. Therefore, from my perspective, I feel that this amended work is now suitable to be published as is in NCOMMs.

Response to the general comments from Reviewer 1

We would like to thank this reviewer for his/her positive evaluation of our response letter and revised manuscript.

Reviewer #2 (Remarks to the Author):

Comment 1 from Reviewer 2

The authors have addressed my comments/concerns except for one point in comment 2. In their response to comment 2, the authors write, "There is indeed a propagation band within the light cone, as verified from our bandstructure simulation." Where is this propagation band within the light cone shown? I do not see it in Fig. 2a. Please add a sentence clarifying this point in the manuscript.

Response to comment 1 from Reviewer 2

We agree that the wording in our previous response letter can cause confusion. As the considered leaky modes have a very short lifetime on the surface, their detection with mode solvers is challenging and an attempt to visualize them has limited value. Based on this fact and the helpful note from this referee, we have removed the relevant sentence from the paper. We have highlighted our current wording in the main text using the track changes option in word and believe this wording is clear.

Comment 2 from Reviewer 2

One more minor comment, the authors may want to clarify which harmonic radiates from the grooves. I believe it is the $n=-1$ harmonic rather than the fundamental.

Response to comment 1 from Reviewer 2

The discussion on which harmonic radiates typically comes up on the analysis and design of spoof surface plasmon waveguides that are periodically modulated in space (see e.g. Scientific Reports 6, 29600, 2016). In our case, we do not have such a modulation and we feel that a discussion of the spatial harmonics is less relevant here.

Reviewer #3 (Remarks to the Author):

Comment 1 from Reviewer 3

In my previous report I put forward two arguments that, in my opinion, needed to be addressed in order to substantiate the main claims of the manuscript.

The first one was related to the elimination of the need of spacer layers, as the proposed structure avoids the coupling to SPPs. In their answer the authors provide calculations for the behavior of dipoles placed at 10nm from the surface. But this calculation can only, at best, imply that the spacer layer should be of the order of 10nm. Clearly dipoles closer to the metal surface than this will present a larger coupling to lossy waves, thus being most probably quenched. So, I do not find the author's reply conclusive.

Response to comment 1 from Reviewer 3

We appreciated the helpful suggestion from this reviewer to add a discussion on lossy waves and believe our arguments and simulations that describe how the creation of a high impedance metasurface can reduce both lossy wave excitation and SPP excitation have definitely improved our paper. The new supplementary Fig. 2 and related text explained the key physics behind all of the relevant optical loss processes for light emitting molecules when they are placed in several high-symmetry locations on the patterned metasurface and at a representative distance (i.e. where SPP and lossy wave excitation are important). The most interesting finding was that the creation of a single groove below an emitter reduces the lossy wave excitation, but does not significantly enhance the emission into free space. The reduced coupling to lossy waves should be expected as the emitter-metal distance is increased upon the creation of the groove. However, the absence of a notable increase in the free-space emission is non-trivial and can be explained by the fact that now more light ends up being coupled to SPP modes due to the larger contribution to the LDOS from SPP modes than free-space modes at a groove opening. For this reason, a single groove is not enough and it is essential to create an array of grooves to efficiently decouple the generated SPPs to farfield radiation. The 10 nm dipole distance was chosen based on our device structure, which has a 15-nm-thick layer of emissive material and 5-nm-thick spacer layer. As we understand, the Reviewer would also like to see what happens for the smallest spacing of the emitters to the high-impedance metasurface. We agree with this reviewer that by moving dipoles closer to the metallic substrate, the lossy wave contribution tends to increase and this makes it harder to get light out of the emitting layer. The results for this dipole-mirror spacing are summarized in the table below.

Dipole position	Loss ratio
Dipole 5 nm above a flat mirror	79%
Dipole 5 nm above a single groove	69%
Dipole 5 nm above a groove array (center of groove)	12%
Dipole 5 nm above a groove array (center of metallic tooth)	60%

From these results it is clear that, if we dig a single groove below the dipole the total emission loss will not change much and a drop in the loss ratio from 79% to 69% is observed. As for the larger spacings, the real benefit to enhancing the emission comes from creating a groove array that can decouple the SPPs to free-space. With the groove array present, the loss ratio drops to just 12%. In the case an emitter is placed above the center of a metallic tooth, the benefits of creating a grooved (HIM) surface are much smaller, as expected. In this case the loss ratio from 79% for the smooth metal to 60% for the HIM. However, from the above discussion it is clear that in all considered cases, the creation of a grooved HIM surface is beneficial for the emission as lossy wave coupling is reduced and the generated SPPs are decoupled very effectively. Similar physics is observed for all emitter spacings. In assessing the overall benefits, it is also worth noting that in our experiment more emissive material is located in the 10-20 nm distance range from the mirror than to what is within the 5-10nm range,

Comment 2 from Reviewer 3

The second comment was about surprise that dimples of this size can also be used to create a high-impedance mirror. This is due not only to the smaller electromagnetic cross-section of dimples, but also on the EM mode structure inside them. If the metal were a perfect electrical conductor, holes with diameters of 130nm would not support propagating modes at wavelengths of 560nm, so their behavior should be very different to that of slits. It could be that the finite dielectric constant of silver would create a propagating

mode, but I doubt it, as the induced change in propagation constant should be huge. In any case, to probe that the proposed metasurface behaves as the authors claim, the authors should have provided a study of the electromagnetic modes, and the associated effective impedances.

So, in my opinion, the authors have not addressed my comments in a satisfactory way. Therefore, I maintain my recommendation of not publishing this manuscript in a high-quality journal as Nat. Comm.

Response to comment 2 from Reviewer 3

We would like to note that this reviewer may have overlooked that the dimples are filled with SiO₂ (not air). This makes it easier for light to penetrate into the dimples and to observe a substantial reflection phase. To make sure that we were correct in our assumptions, we have now simulated the reflection phase for a normally-incident light wave at visible frequencies from a periodic array of dimples with the same dimensions as used to calculate the band structure in supplementary Fig. 7 and added this plot to the figure. As it can be seen from the new supplementary Fig. 7b, the reflection phase changes from -0.8π to 0.8π as conclusive proof that there is a high impedance surface regime for the dimple array. The fields are also seen to nicely penetrate into the dimple as shown in the simulated field profile shown in the inset. This is consistent with the observations made in reference 16 of our paper that discusses RF-based high impedance surfaces.

Reviewers' comments:

Reviewer #2 (Remarks to the Author):

The authors have addressed my concerns.

Reviewer #3 (Remarks to the Author):

Already from my first report, my main issue has been related to the following sentences in the manuscript: "Currently, optical energy dissipation in metallic electrodes is minimized by inserting a dielectric spacer layer between the metal and emitter layers. However, the use of such spacers increases the device's contact resistance and the probability of trapping light inside the high-index light emitting materials via total internal reflection" and "In light of these drawbacks, it would be highly desirable to tackle the dissipative loss in the metal at its root and remove the coupling of emitters to SPPs on the metallic contact without the need for spacer layers and complex light-extraction schemes."

Spacers layers are required to minimize non-radiative losses into lossy waves and, only to a minor extend, to plasmons. As at distances smaller than 20nm dipoles mainly radiate into lossy waves, the loss into plasmons at these distances is almost irrelevant. The previous sentences give the, in my opinion, misleading impression that by eliminating coupling into SPP, the spacer layers are not needed.

In my understanding, the manuscript studies a periodic structure that convert confined plasmons into lossy waves. This is relevant when dipoles radiate mainly into plasmons, i.e, for dipole-metal distances larger than 20nm. For dipoles closer to the metal than this, the proposed structure reduces coupling into lossy waves by a factor of the order of the opening filling factor (as dipoles sitting on top of the opening do not couple to lossy waves). I doubt this reduction is enough not to require a spacer layer.

The authors have presented a full range of calculations, but I do not see a definite answer to this simple point. However, at this stage, and given that all other referees are satisfied with the authors' answers, I believe it is best to present the work to the community via publication in Nat. Comm., and leave readers to judge it (perhaps after consideration by the authors of rephrasing the cited sentences).

Response to the Reviewers' comments:

Reviewer #2 (Remarks to the Author):

Comment 1: The authors have addressed my concerns.

Response to Comment 1:

We thank this reviewer for his/her valuable comments and are glad that he/she is satisfied with our responses.

Reviewer #3 (Remarks to the Author):

Comment 1:

Already from my first report, my main issue has been related to the following sentences in the manuscript: "Currently, optical energy dissipation in metallic electrodes is minimized by inserting a dielectric spacer layer between the metal and emitter layers. However, the use of such spacers increases the device's contact resistance and the probability of trapping light inside the high-index light emitting materials via total internal reflection" and "In light of these drawbacks, it would be highly desirable to tackle the dissipative loss in the metal at its root and remove the coupling of emitters to SPPs on the metallic contact without the need for spacer layers and complex light-extraction schemes."

Spacers layers are required to minimize non-radiative losses into lossy waves and, only to a minor extend, to plasmons. As at distances smaller than 20nm dipoles mainly radiate into lossy waves, the loss into plasmons at these distances is almost irrelevant. The previous sentences give the, in my opinion, misleading impression that by eliminating coupling into SPP, the spacer layers are not needed.

In my understanding, the manuscript studies a periodic structure that convert confined plasmons into lossy waves. This is relevant when dipoles radiate mainly into plasmons, i.e, for dipole-metal distances larger than 20nm. For dipoles closer to the metal than this, the proposed structure reduces coupling into lossy waves by a factor of the order of the opening filling factor (as dipoles sitting on top of the opening do not couple to lossy waves). I doubt this reduction is enough not to require a spacer layer.

The authors have presented a full range of calculations, but I do not see a definite answer to this simple point. However, at this stage, and given that all other referees are satisfied with the authors' answers, I believe it is best to present the work to the community via publication in Nat. Comm., and leave readers to judge it (perhaps after consideration by the authors of rephrasing the cited sentences).

Response to comment 1:

Reviewer 3 is still concerned that one of his/her points remains unanswered. This point pertains to the question whether the losses to surface plasmon polaritons (SPPs) or lossy waves (LWs) in our structures dominate. The Reviewer suspects that the losses to SPPs are insignificant for the thin (15 nm) emitter layers used in this study. In that case, our extensive discussion of SPP excitations seems somewhat overdone/irrelevant. To address this point, we now provide a simple, quantitative calculation of the different loss mechanisms at different dipole-mirror separations. This calculation is presented in supplementary Fig. 1. The calculation clearly shows that the losses to SPPs and LW are both very important. We now mention this upfront in the abstract and

introduction of the main text. We have also rearranged the text a bit to discuss SPPs and LWs on a more equal footing. The valuable input from Reviewer 3 certainly improved the paper, which now provides a more balanced description of both loss mechanisms and how each of them is reduced by patterning the typically smooth metallic electrodes into a high impedance metasurface.

We highlighted all of the changes in the manuscript text file and in the supplementary information.

REVIEWERS' COMMENTS:

Reviewer #3 (Remarks to the Author):

The authors have addressed my comments in a satisfactory way.

I recommend publication of the manuscript.